# The Subjective Experiences of Driving Cessation and Life Satisfaction

**DOI:** 10.3390/bs13100868

**Published:** 2023-10-23

**Authors:** Young-Sun Kim, Hyeri Shin, Sarang Um

**Affiliations:** Department of Gerontology, AgeTech-Service Convergence Major, Graduate School of East-West Medicine Science, Kyung Hee University, Yongin 17104, Republic of Korea; zisoa@khu.ac.kr (H.S.); umlove@khu.ac.kr (S.U.)

**Keywords:** older adults, driving cessation, life satisfaction, mobility

## Abstract

Compared to the driving group, the driving cessation group in this study was found to be a high-risk population in terms of their life satisfaction. This study evaluated data from 315 older adults, aged 55 or older, using the 2018 Korean Older Adults Driving and Mobility Service Trend Survey. These data were collected from 17 representative cities and provinces in South Korea. To minimize the potential for selection bias and the confounding factors inherent in observational studies, this study employed the propensity score matching (PSM) method. Following the matching, multivariate regression analyses were conducted to compare the driving cessation group (*n* = 65) with the driving group (*n* = 50) in terms of their life satisfaction. After adjusting for demographic and health-related variables, the older adults who had ceased driving were found to have lower life satisfaction (Coef. = −1.39, *p*-value = 0.018). Our results highlight the importance of establishing preliminary evidence to guide the development of tailored programs for older adults—especially for those likely to experience diminished life satisfaction and heightened risk—to address the mobility challenges stemming from driving cessation.

## 1. Introduction

Driving is the primary means of mobility, integral to independent living, community engagement, and resource accessibility [1,2]. It underpins essential aspects of daily life, including healthcare access, and augments life satisfaction through the facilitation of social interactions and the provision of independence [3]. Nevertheless, since driving is a behavior that requires multidimensional abilities, such as physical and cognitive abilities [4,5,6], older adults with deteriorating physical functions are more stressed about driving behavior and can easily experience cessation of driving [7,8]. In other words, to drive safely, it is necessary to have visuospatial abilities, such as securing a safe driving distance, and various cognitive judgments, such as responding quickly to emergencies and making appropriate judgments about various driving situations [9,10,11]. However, older adults find it easy to stop driving because their physical and cognitive abilities, such as visuospatial abilities, reaction speed, decision making, and situational judgment, decrease due to aging, which reduces their driving ability [9,10,11]. This driving cessation in older adults imposes limitations on their freedom of movement, substantially diminishing their mobility [12,13]. Therefore, when older adults cease driving, their free movement is restricted, and their mobility is significantly reduced [14,15]. A public transportation environment that can replace driving must be established to maintain the mobility that decreases due to driving cessation. However, for older adults, especially those with disabilities, it is insufficient to replace driving due to the poor quality of the public transportation system, including issues such as difficulties in accessing public transportation and affordability [16,17,18]. Moreover, maintaining driving in older adults means continuing to participate in social life with independence and autonomy. In this respect, stopping driving can negatively affect mental health due to fear of isolation [19]. In particular, as Korea is expected to become a super-aged society, the number of older adults and the number of people who have stopped driving is also increasing [20]. In addition, as each city is implementing a policy of asking older drivers to voluntarily return their driver’s licenses, the number of people subject to driving cessation is expected to continue to increase [21]. Consequently, it is necessary to research this expanding group of individuals who have stopped driving.

Activity theory proposes that older adults can lead a fulfilling later life when they sustain an active lifestyle. It explains that declining activities as a result of aging processes such as retirement, loss of societal roles, and the cessation of driving can lead to diminished life satisfaction during old age [22,23]. Therefore, preserving activity in later life results in beneficial psychological, emotional, and social effects, ultimately leading to improved life satisfaction [22,24]. Prior research has been grounded in activity theory and has found that forms of active engagement in old age, including in activities such as driving [25,26], physical exercise [27], and social engagement [23,28], can be key contributors to life satisfaction. Consequently, it has become necessary to explore the association between driving and life satisfaction, acknowledging it as a vital element in sustaining activity during old age.

Previous studies on driving cessation in older adults have focused on social health, such as social activities and social support [29,30], and mental health, specifically depression [30,31,32]. These studies have confirmed that the cessation of driving triggers a decrease in social interaction and support and intensifies symptoms of depression [29,30,31,32]. Nevertheless, it is crucial to note that when social and mental health deteriorates due to driving cessation, life satisfaction is invariably affected. As a result, the number of recent studies aiming to identify the relationship between driving cessation and life satisfaction are increasing, but they are still insufficient [3,14,15,26].

Upon examining past research, Musselwhite and Haddad (2010) performed focus interviews with older adults, including those who were still driving and others who had recently ceased to do so. As a result, it was confirmed that quality of life deteriorated when older adults gave up driving [3]. Similarly, Liddle et al. (2012) conducted a comparative analysis of the life satisfaction between older adult drivers and non-drivers. They discovered that individuals who discontinued driving exhibited lower life satisfaction than their driving counterparts [14]. Furthermore, a study by Jones et al. (2020) indicated that 18.2% of older adult drivers had reduced their driving during the year prior to their study, with this reduction in driving having a negative correlation with life satisfaction [15]. Moreover, a study by Suntai et al. (2023) investigated driving as a symbol of independence in older adults and its relationship with life satisfaction. The study indicated that individuals who abstain from driving have the lowest well-being, whereas daily drivers are most likely to exhibit the highest well-being [26]. These studies underline the significant positive influence of driving in old age on life satisfaction.

Conversely, in some instances, research studies have reported no observable association between driving and life satisfaction [33,34]. For instance, Peltonen (2013) investigated the relationship between the cessation of driving and life satisfaction, concluding that there was no statistically significant difference in life satisfaction between drivers and non-drivers [33]. Additionally, Schryer et al. (2019) explored the shifts in life satisfaction based on driving status and found that driving cessation was not related to changes in life satisfaction [34].

However, the majority of prior studies confirmed that the cessation of driving negatively impacts life satisfaction [3,14,15,26]. Nonetheless, it is critical to consider that these prior studies had some limitations. For instance, some studies were characterized by a minimal sample size of fewer than sixty participants [3,14], and other studies showed methodological limitations in the generalization of their research results, such as surveying only a few regions [15] or surveying subjects who participated in a specific project [26].

As a result, the objective of this study was to explore the association between driving cessation and life satisfaction in older adults, underscoring the increasing academic and practical significance of driving cessation. Moreover, this study took the divergent findings of past research and methodological limitations into account. Furthermore, this research also used data from older adults residing in diverse local communities across the country. We employed the propensity score matching analysis method to match drivers and those who had stopped driving based on similar characteristics to enable us to more effectively investigate the effect of driving cessation on life satisfaction.

## 2. Materials and Methods

### 2.1. Study Population

Our study used data derived from the 2018 Korean Older Adults Driving and Mobility Service Trend Survey, which were collected by the Department of Gerontology, KyungHee University. The goal of the survey was to identify the mobility patterns and unmet requirements of older adults, both drivers and non-drivers. Accordingly, this survey provided a deep understanding of older adults’ mobility and offers a multidimensional investigation into their physical, mental, and social health in relation to their mobility.

The survey targeted individuals aged 55 years and older, encompassing a nationwide spectrum. Stratified random cluster sampling was employed to collect data from 17 representative cities and provinces across South Korea. The survey was outsourced to a specialized firm, and professional investigators personally conducted face-to-face interviews at each household from December 2018 to January 2019, which collected datasets. F tests-Linear Regression Analysis was conducted using the G-Power program to determine the sample size. In this study, according to the standards of Cohen (1988), the effect size was set to 0.15 (medium), the significance level was set to 0.05, and the power was set to 0.95 [35]. As a result, the total sample size was 446 people, and considering approximately 10% missing samples, data from a total of 501 people were collected.

The final analysis of this study only included 315 individuals, as we excluded those who had never driven a car for our research purposes. 

The ethical approval for the study protocol was granted by the Institutional Review Board of Kyung Hee University (approval number: KHSIRB-18-037).

### 2.2. Measurement

The significant variables in this research were selected based on prior studies. As an independent variable, driving cessation was evaluated by asking the participants, “Are you currently operating a vehicle?” Individuals who responded with, “I am currently driving” were categorized as older adult drivers, while those who answered, “I used to drive, but no longer do so” were classified as individuals who had ceased driving. 

Life satisfaction, as a dependent variable, was measured using the satisfaction with life scale (SWLS) proposed by Diener [36]. The SWLS comprises five questions, with responses originally scored on a seven-point scale. As our study considered respondents who were older adults [37,38], we converted the scale to a five-point scale, based on previous studies (1 = strongly disagree, 5 = strongly agree). The SWLS scale measures subjective life satisfaction in a self-report format. The average value of the responses was calculated and used for the analysis, with higher scores indicating greater subjective life satisfaction. The Cronbach’s alpha of life satisfaction was 0.8322. 

We included the following sociodemographic characteristics as independent variables: gender (1 = male, 0 = female); age; residence (1 = rural, 2 = medium-sized, and 3 = large-sized urban); educational attainment (1 = no formal education, 2 = elementary school, 3 = middle school, 4 = high school, and 5 = college or above); working status (1 = working, 0 = no working); living arrangements (living alone or not); household income; and subjective physical health (1 = healthy, 0 = not healthy).

### 2.3. Statistical Analysis

The main purpose of this study was to investigate the differences in life satisfaction between the group that had ceased driving and the group that were still driving. Given that the group that had ceased driving may have been demographically more vulnerable compared to the driving group, we employed the propensity score matching (PSM) method to adjust for potential selection and confounding biases (e.g., demographic characteristics associated with both groups) [39]. The PSM method is renowned for reducing these biases [40], thereby facilitating a fairer comparison between groups and enhancing the internal validity of the study findings. The versatility of PSM has expanded the scope of research opportunities as it aids in analyzing real-world data—non-randomized data—with diminished selection bias [41].

Using the propensity score is not the only way to correct for the influence of covariates on the treatment effect. The commonly used analysis of covariance (ANCOVA) is another method that can be used for this task. Covariance analysis is effective when the covariate distributions of the treatment and comparison groups are similar. However, when covariate distributions are different, such as in observational studies, the propensity score method is preferred [42,43]. Although finding a matching study with driving cessation in older adults is difficult, the use of the PSM could be found in cross-sectional studies when covariates differed between groups in Korean older adults [44,45]. Our study also found significant differences between the groups that stopped driving and those that did not. Accordingly, we would like to attempt to expand the methodology to driving cessation by adjusting covariates. 

The matching process using the propensity score method is as follows [46,47]. First, variables that affect the possibility of belonging to the experimental group are set as covariates, and it is checked whether there is a significant difference between the experimental group and the control group in the data to be analyzed. Second, a logistic regression analysis is performed with treatment status as the dependent variable and covariates as independent variables, and the probability of being assigned to the treatment group is calculated and used as a propensity score. Third, combined sampling is performed using the propensity score, and cases with the same or similar propensity scores among the cases belonging to the experimental group and the control group are extracted and matched. Fourth, after combined sampling, it is checked whether the experimental group and the control group have equivalent characteristics in terms of covariates. If equivalence is secured, an analysis using the propensity score is performed to estimate the effect of the pure independent variable [48]. 

In order to identify the propensity-score-matched pairs, we performed nearest-neighbor matching, employing the Stata module of psmatch2 [49]. Control variables in our study included gender (1 = male, 0 = female), age, residence (1 = rural, 2 = medium-sized, and 3 = large-sized urban), subway (1 = there is a subway nearby, 0 = no subway), educational attainment (1 = no formal education, 2 = elementary school, 3 = middle school, 4 = high school, and 5 = college or above), working status (1 = working, 0 = not working), marital status (1 = having a spouse, 0 = no spouse), living arrangement (1 = living alone, 0 = living with others), household income, frailty status (0 = robust, 1 = pre-frail, 2 = frail), and subjective physical health (1 = healthy, 0 = not healthy). A total of sixty-five cases of driving cessation were matched with fifty cases of non-driving cessation. To examine the differences between the matched groups after PSM, we conducted T-test and regression analyses after deleting the unmatched cases. Lastly, we performed the sensitivity analysis to conduct the regression analyses with the full sample, including the unmatched cases.

Figure 1 illustrates the unmatched and matched samples. We then compared the groups with driving cessation and drivers in the full study sample (prior to PSM; *n* = 315) and the PS-matched sample (*n* = 115). To identify significant group disparities, we applied Chi-square tests for categorical variables and *t*-tests and ANOVA for continuous variables. Lastly, we performed multivariate regression analyses on the PSM-matched sample to investigate the correlation between driving cessation and life satisfaction. All of the analyses were conducted by using Stata software, version 17.0 (Stata Corporation, College Station, TX, USA).

## 3. Results

### 3.1. Sample Characteristics

The demographics of the respondents are listed in Table 1. From the full study sample, significant differences were noticeable between the driving and the driving cessation group in areas such as gender (*p* < 0.001), age (*p* < 0.01), and household income (*p* < 0.05). Nonetheless, no significant variance was noted between the two groups after matching, thereby verifying the successful execution of the matching. 

The survey respondents’ general characteristics were as follows: In the entire sample, a substantial majority of drivers were male (71.6%); however, of those who had ceased driving, slightly fewer than half were male (55.4%). In the matched sample, the proportion of male drivers was marginally lower, just over half (52%). The average age for the entire sample was 61.9 years for active drivers and 63.89 years for those who had stopped driving. The average age for drivers in the matched sample was 63.3 years, which was similar in the driving cessation group. In terms of residence, the majority resided in urban areas of a small-to-medium size. A higher proportion of current drivers were living in rural areas (22.4% of the full sample and 12% of the matched sample) compared to those who had ceased driving (9.2%). This discrepancy suggests that rural areas may necessitate more driving. Regarding education attainment, the majority had completed high school. The proportion of older adults living alone was higher among those who were in the driving cessation group (15.4%) compared to the current driver group (8.8%). In the matched sample, 14.0% of drivers lived alone; this was adjusted to align closely with those who had ceased driving. The majority of older adults remained employed, with a greater percentage found among the drivers (91.2% of the full sample and 78% of the matched sample). In terms of household income, drivers had higher average income (KRW 3,523,700: USD 2734.52) in comparison to those who had ceased driving (KRW 2,667,700: USD 2069.27). This pattern of higher income for drivers persisted in the matched sample as well (KRW 2,667,700: USD 2177.63).

### 3.2. Relationship between Driving Cessation and Life Satisfaction

Table 2 presents the differences in life satisfaction between the non-driving and driving groups using the matched sample. On a scale of 1 to 5, the overall life satisfaction scores averaged at 16.22 (SD = 3.21) for the drivers and at 14.83 (SD = 3.11) for the non-driving group. The difference was statistically significant (*p*-value = 0.018). In addition, older adults who ceased driving reported lower scores in all aspects of life satisfaction. The aspect that showed the largest difference between the two groups was ‘In most ways my life is close to my ideal’ (driving group mean: 3.22, non-driving mean: 2.87, *p*-value = 0.0319), and the aspect with the smallest difference was ‘The conditions of my life are excellent’ (driving mean: 3.24, non-driving group mean: 3.01, *p*-value = 0.0986). In other words, driving cessation itself might not greatly affect the quality of an older person’s life, but it is presumed to have an impact on their ideal living conditions.

Table 3 presents the results of the regression analysis, which compared the driving cessation and current driving groups within the post-propensity score matching (PSM) sample (*n* = 115). Initially, a univariate analysis was carried out on the relationship between driving cessation and life satisfaction. To reduce the potential confounding effects of the independent variables, a multivariate regression analysis was subsequently conducted with the sociodemographic variables.

From the univariate regression analysis, older adults who ceased driving had lower life satisfaction than those who were still driving (Coef. = −1.39, *p*-value = 0.018). After adjusting for demographic factors, older adults who ceased driving still had lower life satisfaction than those who were still driving (Coef. = −1.45, *p*-value = 0.014). Concerning the covariates, the life satisfaction of older adults was significantly associated with subjective health (Coef. = 1.40, *p*-value = 0.024). 

### 3.3. Sensitivity Analyses

We also examined the association between the non-driving and driving groups among the full study sample. As presented in Table 4, the findings remained similar to the propensity-matched sample. Older adults who had ceased driving had lower life satisfaction than those who were still driving (Coef. = −1.39, *p*-value = 0.018). As shown by the univariate regression analysis, the older adults who had ceased driving had lower life satisfaction than those who were still driving (Coef. = −0.97, *p*-value = 0.021). After adjusting for demographic factors, the older adults who had ceased driving still had lower life satisfaction than those who were still driving (Coef. = −0.88, *p*-value = 0.044). Concerning the covariates, the life satisfaction of the older adults in our sample was significantly associated with their residence (Coef. = 1.05, *p*-value = 0.031); living arrangements (Coef. = −1.36, *p*-value = 0.021); household incomes (Coef. = 0.00, *p*-value = 0.033); and subjective health (Coef. = 1.09, *p*-value = 0.006). 

## 4. Discussion

This study aimed to compare the life satisfaction of older adults who had ceased to drive and those who were still driving. The goal was to examine the impact of driving cessation on the life changes experienced by older adults. Given that the act of driving cessation can be prompted by various circumstances, there may be inherent differences between those who have stopped driving and those who have not. To control the confounding effects, the analysis was conducted using a group of older adult drivers who had stopped driving and who were matched to older drivers with similar demographic variables. 

The differences between previous studies and our study are as follows. First, previous studies on driving cessation in older adults have focused on social health, such as social activities and social support [29,30], and mental health, specifically depression [31,32,33]. Recent studies identifying the relationship between driving cessation and life satisfaction are increasing, but they are still insufficient [3,14,15,26,33,34]. 

Second, previous studies that examined driving cessation and life satisfaction reported that driving cessation negatively affects life satisfaction [3,14,15,26], but some studies found no effect between the two variables [33,34]. Because the research results of previous studies differ, it is necessary to verify the relationship between the two variables. 

Third, in most studies on driving cessation, the numbers of people who had ceased driving were small (approximately 10–30% or less), while most of them were drivers (approximately 70–90% or more), so comparisons were made with a smaller proportion of people who had ceased driving compared to drivers [14,15,26,33,34]. Accordingly, this study verified the relationship between driving cessation and life satisfaction by reducing the selective convenience of the two groups to be closer to the research results of the experimental design by using the propensity score matching (PSM) method.

Fourth, other studies showed methodological limitations by generalizing their research results, such as surveying only a few regions [15] or surveying subjects who participated in a specific project [26]. However, to overcome these limitations of previous studies, this study secured the sample’s representativeness by using data collected from across the country.

Our methodology involved frequency analysis and group difference analyses such as the *t*-test, ANOVA, chi-square, and multivariate regression analysis. The findings from the analyses were multi-fold. 

First, we revealed that older adults who had ceased to drive experienced a lower level of life satisfaction compared to the group that still drove. This contrast remained valid even when the analysis matched demographically similar groups. Generally, the older adults who had ceased driving showed lower life satisfaction, which was statistically significant. These research results are consistent with previous studies that reported that when older adults stop driving, life satisfaction decreases [3,15], and those who ceased driving had lower life satisfaction than drivers [14,26]. In addition, these older adults also scored lower on the life satisfaction scale in comparison to their envisioned ideal life, according to the SWLS. Although it may seem presumptuous to claim that the inability to live an ideal life stems from not driving, we can cautiously infer that driving cessation indirectly impacts this ability.

Second, we found that demographic and sociological variables also significantly impact life satisfaction. When adjusting the demographic factors based on the matched data, subjective health emerged as the only significant variable that correlated with life satisfaction. These results are consistent with previous research that verified subjective health as a significant factor positively affecting life satisfaction [50,51]. Furthermore, when considering the full study sample, life satisfaction appeared higher for those living in small-to-medium-sized cities, cohabiting with someone, possessing a higher household income, and those who self-reported better health. 

In interpreting these research findings, we put forth two recommendations. First, since ceasing to drive directly impacts life satisfaction, it is important to preemptively offer information and education on this significant life change. This education should encompass the provision of information and consciousness-raising initiatives prior to driving cessation. While measures to prevent traffic accidents involving older adult drivers do exist, such as returning a driver’s license, developing and supporting the utilization of driving assistance devices, and education, Korean local governments tend to lean towards actively encouraging license returns.

While ceasing driving could circumvent the cause of traffic accidents in the short term, the long-term implications on mental health for older adults need to be considered. We need to consider the potential psychological stress and reduced quality of life experienced by older adult drivers due to the decreased mobility that they may encounter after they stop driving. This information is also relevant to older adults who voluntarily refrain from driving without being required to formally return their driver’s license. Therefore, providing information and education about how driving cessation could potentially impact their quality of life is crucial for this demographic.

Second, it is essential to identify groups who may be the most negatively impacted after they stop driving and provide them with priority access to focused psychological and behavioral programs related to driving cessation. 

As demonstrated in Appendix A, although the life satisfaction of female drivers and those who have ceased driving was comparable, male drivers showcased significantly lower life satisfaction in comparison to those who had stopped driving, and this difference was statistically significant. 

Other demographic factors were also seen to play a role: for the relatively young cohort (aged between 55 and 64), there was minimal difference in life satisfaction between drivers and non-drivers. However, for individuals aged 65 and over, the gap between the two groups was noticeably larger and statistically significant. 

Older adults who resided in small- and medium-sized cities also exhibited a statistically significant difference. Likewise, older adults who were economically active showed higher life satisfaction when they were driving. 

Among older adults living alone, no significant difference was observed. However, it could be that this result was a result of the small data sample. Interestingly, the older adults who lived alone and had ceased driving recorded the lowest overall life satisfaction score.

Finally, in terms of household income, we examined the higher and lower groups based on the median income of the participants, which was KRW 3 million. The data indicated that older adults with a household income exceeding KRW 3 million per month had a statistically significantly higher life satisfaction when driving compared to those who had stopped driving. 

Groups that had a lower life satisfaction when they stopped driving may have also experienced high stress levels at the time that they stopped driving. As such, there is a need to provide focused programs to alleviate the impending stress they may experience. Based on our research, it would be best to target males, those over 65 years of age, residents of small- to medium-sized cities, and those who are still working.

The insights gained from this study, which investigated the life satisfaction of older Korean adults in relation to their driving status, are two-fold. 

First, our study holds high generalizability, as it conducted analyses using data collected from across the entire country. This was unlike the previous studies that have been conducted on driving cessation, which were limited to certain regions. 

Second, we established that, even after matching potential confounding effects through propensity score matching, driving cessation led to reduced life satisfaction. The implications of these unique characteristics of the non-driving group were seen to be significant, and our study has contributed to a better understanding of them.

Despite the implications of this study, it had limitations. While propensity score matching was used to examine the effects between the experimental and control groups, a longitudinal analysis was not conducted, thus limiting the cross-sectional data. Consequently, future studies should also consider how older people’s quality of life changes over time as a result of the cessation of driving. 

Second, although this study focused on verifying the relationship between driving cessation and life satisfaction, there is a need to additionally consider variables that potentially intervene in driving cessation that were not examined in this study, such as cognitive reserve.

Third, the cessation of driving among older adults does not occur abruptly but tends to happen gradually. Therefore, we can categorize individuals into those who actively drive, those who drive occasionally while gradually reducing their frequency, and those who have stopped driving altogether. In this study, the question was only focused on whether driving had been entirely discontinued; therefore, there is a chance that those who gradually stopped were included in the active driver’s group. Therefore, follow-up research should distinguish between the different stages in the cessation of driving more meticulously, examining the differences between each group.

## 5. Conclusions

In conclusion, this study highlighted the profound effects of driving cessation on the life satisfaction of older adults within the context of the South Korean population. Our results signify the need for an informed and comprehensive approach to encouraging older adults to stop driving, addressing the potential psychological and sociological implications. Moreover, targeted support systems are necessary for groups identified as more vulnerable to life satisfaction changes due to driving cessation, such as males, those aged over 65, residents of small-to-medium-sized cities, and those who are still economically active.

## Figures and Tables

**Figure 1 behavsci-13-00868-f001:**
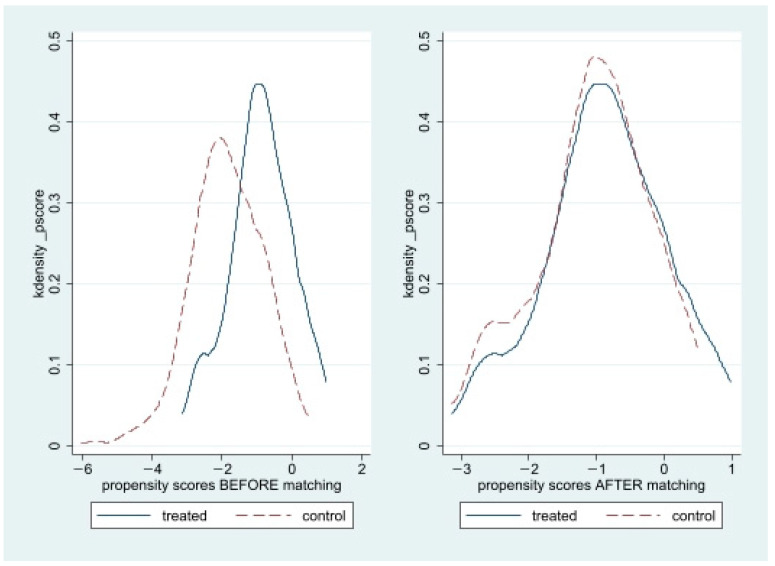
Propensity score graphs: before and after the matching.

**Table 1 behavsci-13-00868-t001:** Sample characteristics of the full study sample and propensity-matched sample.

Groups		Full Study Sample (*n* = 315)			Matched Sample (*n* = 115)		
		Driving (*n* = 250)	Driving Cessation (*n* = 65)	chi^2^/t	Driving (*n* = 50)	Driving Cessation (*n* = 65)	chi^2^/t
		Freq/Mean (SD)	Freq/Mean (SD)		Freq/Mean (SD)	Freq/Mean (SD)	
Gender	Male	179	29	16.75 ***	26	29	0.62
	Female	71	36		24	36	
Age		61.90 (4.84)	63.89 (5.65)	2.85 **	63.3 (5.36)	63.9 (5.65)	−0.57
Residence	Urban	109	30	2.21	21	30	0.31
	medium-sized	85	29		23	29	
	Rural	56	6		6	6	
Educational attainment	Elementary school	21	9	3.95	4	9	1.22
	Middle school	41	15		7	15	
	High school	139	30		32	30	
	College or above	49	11		7	11	
Living arrangement	Living alone	22	10	2.45	7	10	0.04
	Living with others	228	55		43	55	
Work Status	Employed	214	47	6.42 *	39	47	0.48
	Unemployed	36	18		11	18	
Monthly household income ^1^ (KRW/USD)		352.37 (196.66)	266.77 (141.71)	−3.98 ***	280.74 (141.93)	266.77 (141.71)	0.52

Note. Freq—frequency, SD—standard deviation; ^1^ Unit: KRW 10,000 (USD 7.76 in 2023). * *p* < 0.05, ** *p* < 0.01, *** *p* < 0.001

**Table 2 behavsci-13-00868-t002:** Matched group differences in life satisfaction.

Groups	Driving (*n* = 50)		Driving Cessation (*n* = 65)		t	*p*-Value
	Mean	SD	Mean	SD		
Life Satisfaction	16.22	3.02	14.83	3.11	2.4003	0.018
In most ways my life is close to my ideal.	3.22	0.84	2.87	0.83	2.1731	0.0319
The conditions of my life are excellent.	3.24	0.74	3.01	0.69	1.6652	0.0986
I am satisfied with my life.	3.48	0.64	3.18	0.82	2.0823	0.0396
So far, I have gotten the important things I want in life.	3.4	0.69	3.15	0.85	1.6572	0.1003
If I could live my life over, I would change almost nothing.	2.88	0.82	2.6	0.89	1.7175	0.0886

Note: SD—standard deviation.

**Table 3 behavsci-13-00868-t003:** Relationship between driving cessation and life satisfaction (*n* = 115).

	Coef.	S.E.	95% CI		*p*-Value
Main variable ^a^					
Driving cessation	−1.39	0.43	−2.53	−0.24	0.018
Main variable ^b^					
Driving cessation	−1.45	0.58	−2.60	−0.30	0.014
Demographics					
Gender	0.09	0.61	−1.13	1.30	0.887
Age	0.01	0.06	−0.11	0.12	0.932
Residence (ref = rural):					
Medium-sized	0.54	1.00	−1.44	2.53	0.59
Urban	−0.79	1.01	−2.80	1.22	0.44
Education	−0.09	0.37	−0.81	0.64	0.816
Working status (ref = non)	0.14	0.67	−1.19	1.46	0.841
Living arrangement (ref = living alone)	−0.92	0.88	−2.66	0.81	0.294
Household income	0.00	0.00	0.00	0.01	0.429
Subjective health	1.40	0.61	0.19	2.62	0.024

Note: Coef.—coefficient, SE—standard error; ^a^ F(1, 113) = 5.76 (*p*-value = 0.018), R^2^ = 0.0485, adjusted R^2^ = 0.401; ^b^ F(10, 104) = 1.96 (*p*-value = 0.045), R^2^ = 0.1584, adjusted R^2^ = 0.0774.

**Table 4 behavsci-13-00868-t004:** Relationship between driving cessation and life satisfaction (*n* = 315).

	Coef.	SE	95% CI		*p*-Value
Main variable ^a^					
Driving cessation	−0.97	0.42	−1.80	−0.15	0.021
Main variable ^b^					
Driving cessation	−0.88	0.44	−1.74	−0.02	0.044
Demographics:					
Gender	−0.21	0.39	−0.97	0.55	0.582
Age	0.04	0.04	−0.03	0.12	0.248
Residence (ref = rural):					
Medium-sized	1.05	0.48	0.10	2.00	0.031
Urban	0.15	0.48	−0.79	1.09	0.751
Education	−0.09	0.22	−0.53	0.34	0.681
Working status (ref = non)	−0.12	0.46	−1.02	0.79	0.803
Living arrangement (ref = living alone)	−1.36	0.59	−2.51	−0.21	0.021
Household income	0.00	0.00	0.00	0.00	0.033
Subjective health	1.09	0.39	0.32	1.86	0.006

Note: Coef.—coefficient, SE—standard error; ^a^ F(1, 113) = 5.41 (*p*-value = 0.02), R^2^ = 0.0170, adjusted R^2^ = 0.0139; ^b^ F(10, 304) = 3.30 (*p*-value = 0.098), R^2^ = 0.0980, adjusted R^2^ = 0.0684.

## Data Availability

The datasets generated or analyzed in this study are available from the corresponding author upon reasonable request.

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
