# Peer review of "The Subjective Experiences of Driving Cessation and Life Satisfaction"

_behavsci, 2023, doi:10.3390/bs13100868_

Round 1
Reviewer 1 Report
The ms titled “The Subjective experiences of Driving Cessation and Life Satisfaction: A propensity score matching analysis” aimed to explore the association between driving cessation and life satisfaction in a sample of older adults, highlighting the crucial impact of driving cessation on mental health and wellbeing. Results revealed that the older adults who had ceased driving have lower life satisfaction. Au.s discussed results in terms of their practical implications to address challenges posed by the driving cessation.
The ms is well written and the English language is clear. The paper would provide a significant contribution to this research field, highlighting the role of driving cessation in the satisfaction of older adult drivers in various life dimensions. Anyway, in my opinion, some minor changes could contribute to improve the manuscript.
INTRODUCTION:
1) l 31-34: Authors briefly declare that driving cessation is partially determined by the age-related cognitive decline affecting the safe driving performance. This point has been examined in a very general manner, in my opinion. Au.s should explain more in depth those cognitive domain/processes who firstly undergoes to age-related decline preventing the safe driving performance in the elderly. For example, a plethora of studies has focused on reasoning, decision making, and visuospatial abilities (10.1037/0894-4105.18.1.85; 10.1017/S1041610209009119; 10.3390/brainsci11081028). These abilities was found strongly associated to objective measures of fitness-to-drive as well as to self-reported driving behaviours.
2) The introduction encompasses studies on the influence of driving cessation on different dimensions of life satisfaction of the elderly. However, it is important to emphasize the reasons why the elderly is resistant to ceasing their car driving activity. For example, it has been demonstrated that the poor accessibility and affordability of the public transport system are among the key reasons behind such resistance (10.1080/0144164032000048573; 10.3390/su151813972; 10.3390/ijerph182211802). Moreover, remaining able to drive means also to continue to participate in social life with independence and autonomy. This aspect also seems to affect the resistance of the elderly which may be afraid to remain isolated after driving cessation experiencing negative consequences on mental health (10.3390/ijerph20085540).
The addition of these pieces of information can contribute to strengthening the portrait of evidence provided in the introduction section as well as enhancing the informativeness and readability of the paper.
METHODS and RESULTS: Please, provide more detailed descriptions on employed instruments and on methodology.
1) Was a formal power analysis conducted to estimate the minimum sample size? If so, please add this information to the participants' section. If not, please specify which rules of thumb were used to determine the sample size.
2) Please, provide reliability coefficients (Cronbach’s alpha for internal consistency) for each scale of each psychometric standardized tool employed.
3) L 130: I suggest changing the “control variables” in “independent variables”.
4) The table 1 (L. 193) needs to be edited since it is not easily readable. Rows and columns need to be properly aligned.
5) Same for the table 2. I suggest to put the table in a separated page for the sake of clarity.
6) Please, provide parameters on the base of which Au.s performed the matching of participants in the two groups. Moreover, were the participants who did not find a match excluded? Please specify this point.
7) L. 167: Please, remove “<” and “>” around “Table 1”.
DISCUSSION:
1) The discussion lacks a formal connection between the obtained results and the existing literature in this research field. The authors should make an effort to comment on the results in terms of confirmations/disconfirmations in comparison to findings from similar studies.
2) L. 340-344: It may not be a limitation, as, given the necessary sample size to reject the null hypothesis, the authors may discover that they have analyzed a sufficient sample.
3) An additional limitation could relate to potentially intervening variables not examined in the present study (e.g., the cognitive reserve).
The ms is well written and the English language is clear.
Author Response
Thank you very much for devoting your valuable time and sharing your good comments for developing this article.
For valuable comments to improve the quality of this article, we have revised the paper.

Reviewer 2 Report
It is interesting that stopping driving among elderly people lowers their sense of well-being. However, if there is already similar previous research, this paper will not have much value. Furthermore, if, as the author argues, the only problem with previous research is sample size and regional bias, I don't see the point in deliberately adopting methods such as reducing sample size using propensity scores.
A propensity score matching is a technique typically used in observational studies that involve a time course. This is because observational studies cannot be randomized like intervention studies. This method is justified because it reduces the influence of attribute differences on the tendency of change in the dependent variable. However, it has the disadvantage of only considering observed covariates and not latent variables. Therefore, opinions among researchers are divided as to whether it is appropriate to use significantly reducing the number of samples. The authors of the paper seem to consider the use of this method to be a strength and uniqueness of the paper. This can be inferred from its use in the title of the paper. However, if my understand is correct, this method is generally not used in cross-sectional analyses, and not only in studies of driving among older adults. If the author believes that it is correct to use this method in cross-sectional analysis and wishes to emphasize its uniqueness, the author should discuss the validity of adopting this method in more detail. For example, the author should find out what kinds of studies have used this method in the past referring to multiple papers to validate the usage of this tool. ANCOVA is a method of controlling for differences in attributes in cross-sectional analysis without reducing the sample size. The author might have been better off using this tool. Why did the author choose propensity score matching instead of ANCOVA?
minor: There are some areas where proofreading is lacking. For example, the following looks like a repetition of the same content to the best of 258 our knowledge, this was the first study to utilize propensity score matching to examine 259 driving session. 260 To the best of our knowledge, this was the first study to deploy propensity score 261 matching to analyze a group of older adults who had stopped driving.
Author Response

(The authors gave the same response as above.)

Round 2
Reviewer 2 Report
It seems that the points I made are largely reflected in the revised version. However, the rationale for using propensity score matching has not been sufficiently discussed. Specifically, what the author wrote in "response 2" should also be stated in the main text. Please explain why you did not use other methods such as ANCOVA, citing Dehejia and Wahba 1999, Rubin 1997, Shin et al., 2021; Shin & Lee, 2018, etc., as you did in your reply to me.
Author Response
Thank you very much for devoting your valuable time and sharing your good comment for developing this article, particularly the research method. To improve the quality of this article, we have revised the paper based on your valuable comment.

Round 3
Reviewer 2 Report
The authors have fixed all the things.